# Informed Correctors for Discrete Diffusion Models

**Yixiu Zhao**
Stanford University
yixiuz@stanford.edu

**Jiaxin Shi**
Google DeepMind
ishijiaxin@gmail.com

**Feng Chen**
Stanford University
fengc@stanford.edu

**Shaul Druckmann**
Stanford University
shauld@stanford.edu

**Lester Mackey**
Microsoft Research New England
lmackey@microsoft.com

**Scott Linderman**
Stanford University
scott.linderman@stanford.edu

## Abstract

Discrete diffusion has emerged as a powerful framework for generative modeling in discrete domains, yet efficiently sampling from these models remains challenging. Existing sampling strategies often struggle to balance computation and sample quality when the number of sampling steps is reduced, even when the model has learned the data distribution well. To address these limitations, we propose a predictor-corrector sampling scheme where the corrector is informed by the diffusion model to more reliably counter the accumulating approximation errors. To further enhance the effectiveness of our informed corrector, we introduce complementary architectural modifications based on hollow transformers and a simple tailored training objective that leverages more training signal. We use a synthetic example to illustrate the failure modes of existing samplers and show how informed correctors alleviate these problems. On the text8 and tokenized ImageNet $256 \times 256$ datasets, our informed corrector consistently produces superior samples with fewer errors or improved FID scores for discrete diffusion models. These results underscore the potential of informed correctors for fast and high-fidelity generation using discrete diffusion. Our code is available at https://github.com/lindermanlab/informed-correctors.

## 1 Introduction

Denoising diffusion models are powerful generative models for high-dimensional data [1–3]. The central idea of diffusion models is to gradually corrupt data into noise using a "noising" or "forward" process and then to train a parameterized model (usually a deep neural network) to learn its time reversal, commonly known as the "denoising" or "backward" process. In continuous domains like image generation, the forward process is usually defined by gradual injection of Gaussian noise and scaling that transform the data distribution into a standard normal. The backward process is then approximated by learning the gradient of the log density (also known as the "score function") of the marginal distribution. To draw samples from a trained model, one first generates Gaussian noise and then simulates the backward process using the score information. Diffusion generative models are currently the dominant framework for image generation, and they can generate high-resolution images with stunning details given prompts from a user [4, 5].

39th Conference on Neural Information Processing Systems (NeurIPS 2025).

Given the success of diffusion models for continuous data, recent work has explored diffusion modeling in discrete domains [6–12]. Discrete diffusion models can be applied to language [11], protein sequences [13, 14], graphs [15], and more. Notably, Campbell et al. [8] developed a general framework for discrete denoising diffusion in continuous time. They formulated the forward and backward processes as continuous time Markov chains (CTMCs), and they learned to predict the distribution over denoised data via an evidence lower bound objective. Concurrent work by Shi et al. [12], Sahoo et al. [16] and Ou et al. [17] focused on discrete diffusion with the absorbing state forward process (a.k.a., masked discrete diffusion), deriving a simple formulation of the ELBO that reduces to a weighted cross-entropy. Shi et al. [12] also noted that this simple objective provides a unifying perspective to previous works [8, 11, 18], all of which are parameterizations and modifications of the same ELBO objective.

Despite conceptual advances in discrete diffusion, the efficiency-accuracy trade-off in simulating the continuous-time backward process still limit their effectiveness. Song et al. [19] proposed using approximate MCMC correctors to fix the discretization error in simulating the backward process. Campbell et al. [8] extended the predictor-corrector sampling to discrete diffusion models and discovered that the corrector steps significantly impact generative performance. In practice however, a clear understanding of how corrector steps contribute to sample quality is still missing, and recent works [11, 12, 17, 16] still use predictor-only samplers for generation.

We show that predictor-corrector schemes can be improved by leveraging different sampling methods. Analyzing the forward-backward corrector used by Campbell et al. [8], we identify its failure mode in masked diffusion and illustrate it with a simple Markov chain modeling task. To address this shortcoming, we propose the informed corrector — an alternative corrector scheme inspired by adaptive Gibbs sampling that fixes the issues of the forward-backward corrector and explicitly targets low-probability dimensions. Sampling with the informed corrector drastically decreases error rate in our synthetic example, as well as in Text8, where the model makes much fewer spelling errors than baselines. On tokenized ImageNet $256 \times 256$, we show that the informed corrector yield diverse, high quality samples competitive with existing diffusion models.

## 2 Background

### 2.1 Continuous-Time Discrete Diffusion Models

Consider a data vector $x_0 \in \mathcal{S}^D$ sampled from a data distribution $p_{\text{data}}$, where $\mathcal{S}$ is the base space for each component. Denoising diffusion models [1–3, 19] model this unknown data distribution by progressively adding noise to data samples until their distribution converges to a simple target, such as a uniform distribution. Then, a deep neural network is trained to invert this process and remove the noise. This reverse process forms a generative model from which new samples can be drawn. The noising and generative processes are also called the *forward* and *backward* processes, respectively. We will adopt this convention for the rest of the paper.

Continuous-time diffusion models define a forward process with noising distribution $q_t(x_t|x_0)$ and marginals $q_t(x_t) = \int q_t(x_t|x_0)q_0(x_0)\mathrm{d}x_0$, with $q_0 = p_{\text{data}}$ and limiting distribution $q_T \approx \pi$. The forward process is designed so that $\pi$ is simple. Then, a parameterized process with marginals $p_t^\theta$ is learned to match the true backward process.

When the data is discrete (e.g., text, protein sequences, neural spike counts), the base space becomes a "vocabulary" $\mathcal{S} = \{1, \ldots, S\}$ of size $S$, and the forward process can be described as a CTMC with initial distribution $q_0$ and transition rate matrix $R_t \in \mathbb{R}^{S^D \times S^D}$. The off-diagonal terms $R_t(x, y)$ of the matrix represent the rate of state $x$ transitioning to state $y$ at time $t$, and the diagonals $R_t(x, x)$ are defined as the negative sum of all other elements in the same row: $R_t(x, x) = -\sum_{y \neq x} R_t(x, y)$. Assuming that $R_s, R_t$ commute for all $s$ and $t$, the finite-time transition probabilities satisfy

$$q_{t+\Delta t|t}(y \mid x) = \left[\exp\left\{\int_t^{t+\Delta t} R_s \, \mathrm{d}s\right\}\right](x, y) = I(x, y) + R_t(x, y)\Delta t + o(\Delta t),$$

where $I$ represents the identity matrix, and in the square brackets we compute a matrix exponential. The time reversal of this CTMC is also a CTMC, and its transition rate is,

$$\tilde{R}_t(y, x) = R_t(x, y)\frac{q_t(x)}{q_t(y)} = R_t(x, y)\sum_{x_0}\frac{q_{t|0}(x \mid x_0)}{q_{t|0}(y \mid x_0)}q_{0|t}(x_0 \mid y). \tag{1}$$

Campbell et al. [8] approximated the backward rate by learning a parameterized denoising model $p_{0|t}^\theta(x_0 \mid y) \approx q_{0|t}(x_0 \mid y)$. Alternatively, Meng et al. [20] and Lou et al. [11] observed that the ratio $s_t(y)_x \triangleq q_t(x)/q_t(y)$ plays a similar role as the continuous score function $\nabla \log q_t(y)$, up to an additive constant. Therefore, they proposed to learn an approximate score function instead, $s_t^\theta(y)$.

The training objective is a negative evidence lower bound (ELBO), which is a function of the approximate backward rate $\tilde{R}_t^\theta(y, x) = R_t(x, y)s_t^\theta(y)_x$:

$$\mathcal{L}(\theta, x_0) = \int_0^T \mathbb{E}_{q_{t|0}(y|x_0)}\Big[\sum_{x \neq y}\big\{\tilde{R}_t^\theta(y, x) - R_t(x, y)\frac{q_{t|0}(x \mid x_0)}{q_{t|0}(y \mid x_0)}\log(\tilde{R}_t^\theta(y, x))\big\}\Big]\,\mathrm{d}t + C, \tag{2}$$

where $C$ is a constant independent of $\theta$.

Given the intractable sums over the exponentially large state space $\mathcal{S}^D$ above, it is natural to constrain the forward process to be a composition of identical independent processes on each dimension, where the rate over the entire state $R_t$ can be written as

$$R_t(x, y) = \beta(t)\sum_{d=1}^D R_b(x^d, y^d)\mathbb{1}_{\{x^{\backslash d}=y^{\backslash d}\}} \tag{3}$$

for a base rate matrix $R_b \in \mathbb{R}^{S \times S}$, absorbing the time dependence into the scalar coefficient $\beta(t)$. Here, $x^{\backslash d}$ represents all components of $x$ apart from the $d$-th dimension. Backward rates are also factorized with this forward rate factorization, allowing the objective in (2) to be nicely decomposed into a sum over dimensions, as detailed by Campbell et al. [8].

## 2.2 Absorbing Discrete Diffusion

For generic discrete data, two forward processes are most commonly used: the uniform process and the absorbing state process [7]. We focus on the absorbing state process in this paper, for which the base rate matrix for each dimension can be written as

$$R_b^{\mathrm{absorb}}(x^d, y^d) = \begin{cases} 1 & y^d = \mathrm{MASK},\ x^d \neq \mathrm{MASK} \\ -1 & y^d = x^d,\ x^d \neq \mathrm{MASK} \\ 0 & \text{otherwise,} \end{cases} \tag{4}$$

where we augment the vocabulary set $\mathcal{S}$ by introducing the MASK token, which represents the absorbing state. The absorbing state diffusion is important to consider because it is very commonly used to model sequence data without ordinal structure (e.g., text and protein sequences). It is also intimately connected to masked language models [7, 21, 22].

We use special notation for the absorbing state process. For a sequence $x \in \{\mathrm{MASK}, 1, \ldots, S\}^D$, we use the mask operator $y = M^d(x)$ to denote the sequence obtained by changing the $d^{\text{th}}$ component of $x$ to MASK. Finally, we denote the masked indices of $x$ by $\mathcal{M}(x) \equiv \{d : x^d = \mathrm{MASK}\}$. The set of non-mask indices is the complement, $\overline{\mathcal{M}}(x)$.

Recently, Shi et al. [12] introduced the MD4 diffusion model, which outperformed previous discrete diffusion models on both text and pixel-level image generation benchmarks. Furthermore, they noted that several continuous time denoising diffusion objectives (including (2) ) are mathematically equivalent and can be greatly simplified in the case of masked diffusion. The simplified MD4 loss is:

$$\mathcal{L}_{\mathcal{M}}(\theta, x_0) \triangleq \int_0^1 \mathbb{E}_{q_{t|0}(x|x_0)}\left[\frac{\alpha_t'}{1 - \alpha_t}\sum_{d \in \mathcal{M}(x)}\log p_{0|t}^\theta(x_0^d \mid x)\right]\,\mathrm{d}t, \tag{5}$$

where $\alpha_t \triangleq \exp\left(-\int_0^t \beta(s)\,ds\right)$ is the survival probability of the token at each position and $\alpha_t'$ indicates its time derivative. We denote this loss as $\mathcal{L}_{\mathcal{M}}$ since it sums over the masked dimensions. As we will show later, this loss has a counterpart, $\mathcal{L}_{\overline{\mathcal{M}}}$, that can be combined with the MD4 loss to yield better results for training our diffusion model.

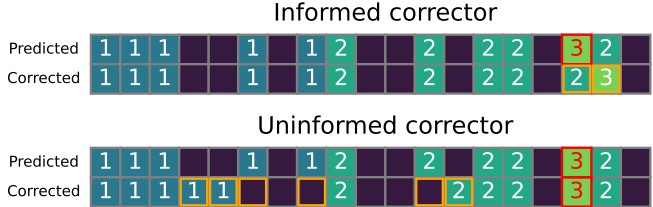

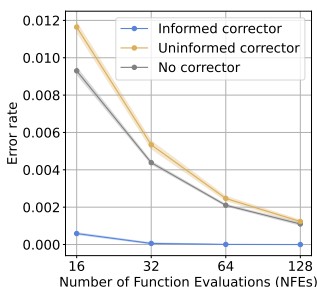

Figure 1: Demonstration of the failure modes of the uninformed corrector on the Markov chain example. The predictor made a mistake (red) in sampling an impossible transition, which is immediately corrected by the informed corrector. The uninformed corrector, however, masks out tokens at random, while unmasking other tokens at the same time. Changes made by the correctors are highlighted in orange.

Figure 2: The informed corrector substantially lowers error rates on the Markov chain synthetic experiment.

## 2.3 Sampling From the Backward Process

Campbell et al. [8] proposed to use tau-leaping, an approximate sampling method that allows simultaneous updates to multiple dimensions for a single model evaluation. Tau-leaping assumes that the backward process transition rate is constant between $[t - \tau, t)$, regardless of whether states change during that interval. Under this assumption, multiple transition events in the interval can be applied simultaneously. Leverging the structure of masked diffusion, MD4 [12] used a simpler approach. Instead of trying to simulate the continuous time process, they directly sampled from the finite-time backward conditional distribution $p^\theta_{t-\Delta t|t}(\,\cdot\mid x_t)$. They call this approach ancestral sampling.

**Corrector steps.** Both the above approaches treat the one-step backward update as independent across dimensions, which accumulates simulation errors over the course of the backward process. To combat this problem, additional corrector steps can be used to ensure the marginal distribution of samples $x_t$ at time $t$ matches $q_t$. Campbell et al. [8] show that a corrector process with rates,

$$R^c_t(x, y) = R_t(x, y) + \tilde{R}_t(x, y) \tag{6}$$

leaves the stationary distribution $q_t$ invariant. In practice, one must use the learned backward rates $\tilde{R}^\theta_t$ as an approximation for the true backward rates $\tilde{R}_t$. This can be thought of as an "uninformed" corrector, as it is simply a summation of the forward and backward rates, neither of which explicitly targets problematic tokens in the masked diffusion setting. Nevertheless, Campbell et al. [8] show that this uninformed corrector increases sample quality in practice.

**Correcting for the absorbing diffusion.** The necessity of corrector steps is particularly apparent when tau-leaping is used in conjunction with an absorbing forward process. In the reversal of the absorbing process, only transitions from the absorbing state to other states are allowed, which is reflected by the structure of $\tilde{R}_t$. Thus, once a token has been generated, it cannot be erased or changed, rendering it impossible for the predictor to fix its own errors. Forward-backward corrector steps mitigate this issue by introducing the forward term $R_t$, allowing transitions in the forward direction, resulting in generated tokens being "masked out" again with uniform probability.

Though forward-backward corrector steps solve the issue in theory, they are far from optimal in practice. The steps can only correct mistakes by masking tokens at random. Ideally, we would like the corrector to make informed transitions based on which states are more likely under the model. This observation motivates us to find a better alternative to the forward-backward corrector.

## 3 Methods

To address the limitation of existing uninformed correctors, we design a corrector scheme that uses the model to inform which tokens to correct, prioritizing tokens that are more likely to have errors. Next, we introduce a hollow transformer-based architecture to parameterize this informed corrector, along with a tailored training objective to enable efficient training of the corrector. We describe these methods in detail below.

## 3.1 Informed Correctors

To design a corrector scheme for our sampler, we first identify a Markov chain that has the marginal distribution $q_t$ as a stationary distribution. For discrete distributions, a straightforward option is to use Gibbs sampling,[1] i.e., we iteratively select an index from $d = 1, \ldots, D$ and resample the $d$-th dimension according to the conditional distribution

$$x^d \sim q_t(\,\cdot\mid x^{\backslash d}). \tag{7}$$

There are two common ways of selecting the index $d$, known as systematic scan and random scan. Systematic scan enumerates the dimensions following a fixed permutation, whereas random scan selects the dimensions uniformly at each resampling step. In both cases, it can take many steps for the Markov chain to "correct" a simulation error in one of the dimensions.

**Informed selection.** To improve efficiency, we use an adaptive random-scan Gibbs sampler with a non-uniform selection distribution [25–28]. We choose this distribution so that the corrector prioritizes updating those dimensions that are more likely to have simulation errors. To achieve this, we let $c_d$ be a confidence score indicating the likelihood of correctness in the $d$-th dimension of the current sample. We select the indices to update in the next $k$ steps $d_1, \ldots, d_k$ by sampling without replacement from the distribution $\mathrm{Cat}(\frac{\exp(-c_d/\tau)}{\sum_{d'} \exp(-c_{d'}/\tau)})$, where $\tau$ is a temperature parameter. This sample can be generated using the Gumbel-top-$k$ trick [29, 30]: First draw $D$ independent Gumbel random variates, $g_d \sim \mathrm{Gumbel}(0,1)$, one for each dimension. Then rank the dimensions in descending order of $-c_d/\tau + g_d$ and select the top $k$ dimensions.

We consider two choices of the confidence score. The first option is $c_d \triangleq \log q_t(x^d \mid x^{\backslash d})$, the log-probability of the $d$-th dimension of the current sample given all other dimensions, where:

$$q_t(x^d \mid x^{\backslash d}) = \begin{cases} 1 - \alpha_t & x^d = \text{MASK} \\ \alpha_t q_{0|t}(x^d \mid x^{\backslash d}) & x^d \neq \text{MASK} \end{cases}, \tag{8}$$

and $q_{0|t}(x^d \mid x^{\backslash d})$ is the denoising conditional distribution. An alternative confidence score is

$$c_d^{\mathrm{margin}} \triangleq \log q_t(x^d \mid x^{\backslash d}) - \max_{i \neq x^d} \log q_t(x^d = i \mid x^{\backslash d}). \tag{9}$$

Similar to in [31], the advantage of this definition is that it maintains low confidence for a dimension when there is an alternative token that the model assigns comparable high probabilities (i.e., $\max_{x'^d \neq x^d} \log q_t(x'^d \mid x^{\backslash d})$ is equally large). In practice, we often use this definition of the confidence as we found it consistently outperforms the former.

**Parallel updates.** Rather than updating the $k$ dimensions sequentially, we update them in parallel, as in Hogwild Gibbs sampling [32]. In practice, we want to keep $k$ small to avoid errors in the parallel resampling, while keeping it large enough so that errors can be fixed reliably. We study the effects of this hyperparameter in section 5.3.

**Conditioning on the mask configuration.** In masked diffusion, a sample $x_t \sim q_t(\cdot)$ contains both mask and non-mask tokens. The Gibbs-inspired update above might turn a mask into a non-mask token or vice versa. This is nonoptimal — on the one hand, the corrector steps are not introduced to generate new tokens but rather to fix errors introduced by the predictor steps; on the other, re-masking tokens that the predictor will later unmask creates redundancy. To improve corrector efficiency, instead of the marginal distribution $q_t(\cdot)$, we target the mask-conditional distribution,

$$q_t(\,\cdot\mid \mathcal{M}(x)) \propto q_t(\cdot)\mathbb{1}_{\mathcal{M}(\cdot) = \mathcal{M}(x)}. \tag{10}$$

In other words, starting from the current sample $x$, we fix the mask configuration $\mathcal{M}(x)$ and only allow the corrector to change non-mask tokens into other non-mask tokens. To update position $d \in \overline{\mathcal{M}}(x)$, we now simply need to sample from the denoising conditional distribution:

$$x^d \sim q_{0|t}(\cdot^d \mid x^{\backslash d}). \tag{11}$$

---

[1]Other Markov chains could be employed instead. For example, the forward-backward corrector proposed by Campbell et al. [8] leverages a birth-death process that has $q_t$ as the stationary distribution [23]. We also tested locally-balanced Markov processes [24], but their performance was inferior to Gibbs.

Intuitively, this is true because a non-mask token at time $t$ must be consistent with the token from the true token at time 0 given the masking forward process. This is convenient for us, since the only distribution required by the informed corrector scheme is $q_{0|t}(x^d \mid x^{\backslash d})$. For derivation and more implementation details, see appendix B.

Next, we will discuss how to implement this corrector scheme efficiently, so that only one network evaluation is needed to select and update positions $d_1, \ldots, d_k$ simultaneously.

## 3.2 Parameterizing the Conditional Distribution Using Hollow Transformers

Since the MASK token provides no information about the original data, we make the further observation that $q_{0|t}(x^d \mid x^{\backslash d}) = q_{0|t}(x^d \mid M^d(x))$, which is the familiar denoising distribution that we can approximate using $p_{0|t}^\theta(x^d \mid M^d(x))$. Therefore, we can implement the corrector on dimension $d$ by calling our denoiser on $M^d(x)$. However, correcting $k$ dimensions in this case requires $k$ forward passes through the denoiser. In order to simultaneously obtain $q_{0|t}(\cdot^d \mid x^{\backslash d})$ for multiple $d$, we need an architecture where information about $x^d$ does not propagate to the output at the $d$th position.

We solve this problem with a hollow transformer [33], a two-stream self-attention architecture where each stream runs causal attention in a different direction. The embeddings are then offset by one position so that the embedding at position $d$ does not attend to the corresponding input position. The two streams are then summed together and sent through multilayer perceptron layers to the output. By construction, the hollow transformer satisfies the property that for all $(t, \theta)$,

$$x^{\backslash d} = y^{\backslash d} \implies f^\theta(x, t)_d = f^\theta(y, t)_d, \tag{12}$$

where $f^\theta(x, t)_d$ represent network output at sequence position $d$ with inputs $(x, t)$ and parameter $\theta$. In other words, the output at dimension $d$ is agnostic to the input at the same dimension $d$. We expand upon architectural details of the hollow transformer in appendix C.

Once we have the hollow transformer, we can use it to parameterize the denoiser, which is learned through regular masked diffusion training. This denoiser can then be used for both the predictor and corrector steps in sampling. Nonetheless, we observe that switching from standard transformer to hollow transformer can sometimes degrade the predictor performance, thereby diminishing the overall benefit of correctors. In these situations, we propose to keep the original denoiser architecture and train a separate hollow transformer for the corrector. We will show this in section 5.3.

## 3.3 Learning the Hollow Transformer with the ELBO

The preceding section demonstrated that the corrector can be trained using the same ELBO objective employed for training the denoiser. Interestingly, we reveal that for hollow transformers, there is an alternative expression of the ELBO that offers training signals that are complementary to the widely used simplified objective (20). This new expression is presented in the following proposition.

**Proposition 3.3.1** *When $R_t = R_t^{absorb}$, the continuous-time objective function* (2) *simplifies to*

$$\mathcal{L}_{\overline{\mathcal{M}}}(\theta, x_0) = \int_0^1 \mathbb{E}_{q_{t|0}(x|x_0)} \left[ \frac{\alpha_t'}{\alpha_t} \sum_{d \in \overline{\mathcal{M}}(x)} \log p_{0|t}^\theta(x_0^d \mid M^d(x)) \right] \mathrm{d}t + C. \tag{13}$$

We include a full derivation of this result in appendix A and show that it is related to $\mathcal{L}_\mathcal{M}$ via a simple label-switching trick. Notably, Campbell et al. [8] and Shi et al. [12] both avoid writing the loss in this form, since evaluating $p_{0|t}^\theta(x^d \mid M^d(x))$ for each different $d$ requires a separate call to the denoising network, which is prohibitively inefficient in practice. The hollow transformer fixes this issue, as now the inner sum over $d$ can be evaluated with one forward pass of the network.

Furthermore, we observe that this different form of the ELBO complements the MD4 loss in (5), in the sense that it sums over the non-mask dimensions $\overline{\mathcal{M}}(x)$, while MD4 sums over the masked dimensions $\mathcal{M}(x)$ instead. Both losses ignore half of the learning signal from the sample $x \sim q_{t|0}(\cdot \mid x_0)$, which seems unideal. This observation suggests that a more informative objective can be obtained by averaging the two losses:

$$\mathcal{L}_{\text{HD}} = \tfrac{1}{2}(\mathcal{L}_{\overline{\mathcal{M}}} + \mathcal{L}_\mathcal{M}). \tag{14}$$

Since both terms are different expressions of the same ELBO, $\mathcal{L}_{\text{HD}}$ can be viewed as applying a variance reduction trick for the sample estimator. We observe in experiments that this objective indeed outperforms its individual components and makes training converge faster (see appendix D.2). We term our hollow transformer masked diffusion model trained with loss (14) **HollowDiff**.

## 4 Related Work

**MaskGIT.** Alongside the development of discrete diffusion models, other classes of non-autoregressive models employing similar iterative generation principles have emerged [34, 35, 33]. Among these, MaskGIT [35] is notably similar to masked diffusion models [12, 36], particularly as both are trained using denoising cross-entropy losses. However, key differences distinguish them: MaskGIT utilizes a non-likelihood-based weighting for its loss function and employs a confidence-based decoding order. In contrast, masked diffusion models typically unmask tokens in a random order that aligns with their theoretical backward process. Inspired by MaskGIT, large language diffusion models such as LLaDA [37] adopt similar heuristics for remasking tokens at each prediction step. This effectively alters the predictor's unmasking order based on confidence, which can introduce bias into the sampling process, causing it to deviate from the diffusion model's theoretical reverse process. In contrast, our confidence metric is only used to define the selection distribution of a random-scan Gibbs sampler, and the informed corrector preserves the marginal distributions of the original diffusion framework to first order.

**Discrete diffusion correctors.** Campbell et al. [8] and Gat et al. [38] introduced the forward-backward and DFM correctors for discrete diffusion. However, these "uninformed" correctors can only address errors through inefficient random re-masking. ReMDM [39] proposed the remasking diffusion process that includes forward-backward and DFM as special cases, while effectively combining the predictor and corrector steps into one. Relatedly, DPC [40] proposed a corrector scheme for MaskGIT sampling. Although their method can be seen as "informed", they reportedly require hundreds of steps to achieve good sample quality on ImageNet 256×256.

## 5 Experiments

In section 5.1, we test the informed corrector against other sampling methods in a hidden Markov model setting where the true denoising conditional distribution is known. We find that informed correctors consistently reduce error rates across a wide range of function evaluation counts. This observation is confirmed in section 5.2 on real-world data, where informed corrector also greatly reduce spelling errors in sampled text. In section 5.3, we use informed correctors for tokenized image synthesis, achieving competitive FID scores on ImageNet $256 \times 256$ with a 230M parameter model and only a small number of function evaluations.

### 5.1 Hidden Markov Modeling

To check if informed correctors improve sample quality, we propose a simple setting where the true denoising conditional distribution $q_t$ is easily accessible. Consider sequence data $x \in \{0, \ldots, S-1\}^D$ sampled from a Markov chain: $q_0(x) = q_0(x^1) \prod_{d=2}^{D} q_0(x^d \mid x^{d-1})$, where $q_0(x^d = i \mid x^{d-1} = j) = A_{ij}$ and $x^1$ is sampled uniformly. At time $t$ of the forward process, the denoising model observes a masked version of this sequence:

$$q_{t|0}(x_t \mid x_0) = \prod_{d=1}^{D} q_{t|0}(x_t^d \mid x_0^d) = \prod_{d=1}^{D} \left\{ \alpha_t \mathbb{1}_{\{x_t^d = x_0^d\}} + (1 - \alpha_t) \mathbb{1}_{\{x_t^d = \text{MASK}\}} \right\}, \qquad (15)$$

and tries to predict the conditional denoising likelihood $q_{0|t}(x_0^d \mid x_t^{\backslash d})$. This can be seen as performing inference in a hidden Markov model (HMM), where the prior is the Markov chain $q_0$, and the observation model is $q_{t|0}$. Given this, we can perform standard message-passing inference [41] to find the posterior $q_{0|t}(x_0^d \mid x_t^{\backslash d})$.

Table 1: A comparison of guidance-free discrete diffusion and MaskGIT models on ImageNet $256 \times 256$. We report two informed corrector approaches (see section 3.2): "HollowDiff + informed corrector" uses a hollow transformer for both predictor and corrector, while "MD4 + informed corrector" uses a separately trained hollow transformer model to correct predictions made by the MD4 model. We use 8 predictor steps, 8 corrector steps, and 1 final denoising step.

| Method | # Params | # Steps | FID ↓ | IS ↑ | Precision ↑ | Recall ↑ |
|---|---|---|---|---|---|---|
| MaskGIT [42] | 227M | 18 | 6.56 | 203.6 | **0.79** | 0.48 |
| DPC [40] | 391M | 180 | **4.45** | **244.8** | 0.78 | **0.52** |
| VQ-Diffusion [43] | 518M | 100 | 11.89 | - | - | - |
| ReMDM [39] | 227M | 16 | 7.40 | 145.27 | - | - |
| MD4 | 230M | 17 | 6.28 | 175.32 | 0.79 | **0.48** |
| + Uninformed corrector | 230M | 17 | 7.41 | 174.49 | 0.80 | 0.44 |
| + ReMDM sampler | 230M | 17 | 6.51 | **246.34** | **0.85** | 0.40 |
| *Informed Corrector (Ours)* | | | | | | |
| HollowDiff + informed corrector | 230M | 17 | 6.26 | 188.18 | 0.78 | 0.45 |
| MD4 + informed corrector | 400M | 17 | **5.78** | 212.52 | 0.82 | 0.43 |

To demonstrate the problems with existing samplers, we choose a structured transition matrix $A$:

$$A_{ij} = \begin{cases} p & j = i \\ 1 - p & j = (i + 1) \bmod S \\ 0 & \text{otherwise.} \end{cases} \tag{16}$$

At each position, the token $x^d$ has probability $p$ of staying fixed and probability $1 - p$ of changing into the next token. To evaluate sample quality, one important metric is the number of errors in the samples, i.e., occurrences of impossible transitions within the sampled sequence. The overall error rate is computed by:

$$r_{\text{err}}((\hat{x}_n)_{n=1}^N) = \frac{1}{N(D-1)} \sum_{n=1}^{N} \sum_{d=1}^{D-1} \mathbb{1}_{\{\hat{x}_n^{d+1} \neq \hat{x}_n^d\}} \cdot \mathbb{1}_{\{\hat{x}_n^{d+1} \neq (\hat{x}_n^d - 1) \bmod S\}}. \tag{17}$$

Since the samplers have access to the true denoising distribution, the error rate will be 0 if no more than one token is generated in a single step. However, if more than one token is generated during a parallel update, there is a risk of generating impossible transitions.

We show error rates (mean and standard deviation over 5 random seeds) in Fig. 2. We find that ancestral sampling with the informed corrector yields the least error for all examined NFE settings. Meanwhile, the uninformed corrector consistently performs worse than using no corrector at all, showing that an uninformed corrector step is not as valuable as an additional predictor step in this setting (see fig. 1). More experimental details can be found in appendix D.

## 5.2 Character Level Text Modeling

We next evaluate our method on the character-level modeling task using the text8 dataset [44]. Following standard practice, we use the provided dataset splits and train on text chunks of length 256 for 1 million steps using a batch size of 512. As a simple measure of sample quality, we compute the error rate, defined as the fraction of total characters in words that are absent from the training set vocabulary.

We benchmark against MD4 [12], which employs a standard transformer architecture and has a comparable parameter count (fig. 3). HollowDiff significantly outperforms the standard MD4 baseline in MD4 enhanced with the uninformed corrector and the ReMDM sampler, demonstrating the utility of our proposed sampling framework. We direct the readers to table 3 for examples of generated text.

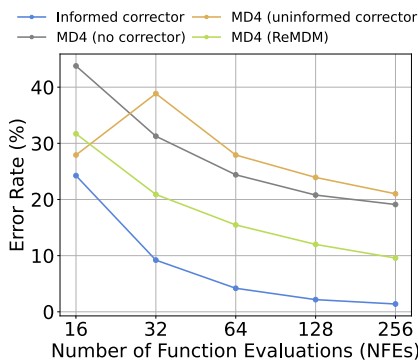

Figure 3: Comparison of word error rate across different sampling methods and models on the text8 dataset.

## 5.3 Class-conditional Image Modeling

We evaluate our method on tokenized ImageNet 256×256 using the VQ tokenizer from [43]. All models are trained on the standard training split and evaluated without classifier-free guidance. Table 1 presents a comparison of generative models in the guidance-free setting. HollowDiff with the informed corrector achieves an FID of 6.26, using only 17 total steps (8 predictor, 8 corrector and a final denoising step). This outperforms several baselines with similar or larger model sizes and substantially more sampling steps. VQ-Diffusion [43] requires 100 steps to reach FID 11.89, while ReMDM [39] achieves FID 7.40 in 16 steps. We also note that the MD4 baseline achieves similarly strong results without any correctors, and the uninformed corrector and ReMDM sampler fail to improve its performance further. Generated samples from our method are shown in appendix D.3.5.

**Transformer architectures.** Without using correctors, HollowDiff significantly underperforms MD4 in FID (10.24 vs 6.28 in table 2), indicating Hollow transformer's subpar denoising capabilities compared to standard transformer on image data. This deficiency is offset by its capability to "self-correct" via informed correctors.

To further demonstrate the value of informed correctors, we also test a hybrid setup where MD4 serves as the predictor and HollowDiff acts as the corrector. This combination improves FID to 5.78, exceeding the performance of both individual models (see table 1 for a full set of metrics). Despite the hybrid setup using more total parameters (400M) than the individual HollowDiff or MD4 model (230M), it highlights the modularity and compatibility of our corrector: it can be applied to stronger pretrained predictors without need for joint training. This finding confirms that our corrector provides complementary gains outside of architectural choice, and is broadly applicable to discrete diffusion models in general.

Table 2: FID scores (ImageNet 256) for MD4 vs HollowDiff under various corrector configurations.

| Model + Corrector | FID $\downarrow$ |
|---|---|
| HollowDiff (no corrector) | 10.24 |
| HollowDiff + uninformed corrector | 10.93 |
| HollowDiff + informed corrector | 6.26 |
| MD4 (no corrector) | 6.28 |
| MD4 + informed corrector | **5.78** |

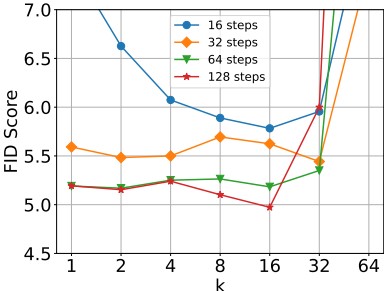

Figure 4: FID vs. number of informed corrector updates $k$ under various sampling budgets. Increasing $k$ initially improves efficiency but degrades sample quality for large $k$.

**Number of parallel updates.** To investigate how the number of parallel updates affect sample quality, we conduct an ablation test on $k$ with the hybrid MD4 + informed corrector setting. Figure 4 shows the FID scores as a function of $k$, across different total number of update steps (evenly split between predictors and correctors). For each number of update steps $\{16, 32, 64, 128\}$, we find that one temperature parameter $\tau = \{1.0, 2.0, 5.0, 10.0\}$ respectively works best across all values of $k$.

We observe that for small NFE budgets, increasing $k$ significantly improves sample quality up to a point. For higher NFE budgets, performance is much less sensitive to $k$, and a small value (e.g., $k$=1 or $k$=2) already yields near-optimal FID. This aligns with the intuition that predictor steps dominate performance at high compute budgets. When too many dimensions are updated simultaneously, their mutual dependencies are ignored, leading to update collisions where corrections can contradict or undo each other. This is observed empirically as in all cases, too large values of $k$ (e.g., $k \geq 32$) lead to worse FID.

## 6 Discussion

We proposed informed correctors and an accompanying training framework for learning and sampling from masked discrete diffusion models. We showed that the uninformed forward-backward corrector is not well-suited to the masked diffusion setting, since it does not leverage information about the conditional likelihood of the generated tokens. Our approach, HollowDiff, produces superior sample quality with fewer model evaluations than alternative discrete diffusion samplers both in the synthetic setting and real-world datasets. Overall, our work serves to further enhance the capabilities

of diffusion models in discrete regimes and to bridge the sampling quality gap between discrete diffusion models and other types of generative models.

**Limitations and future work.**  Using the informed correctors for absorbing diffusion requires the use of the hollow transformer architecture. As a result, we cannot apply the correctors directly to arbitrary pretrained models. Learning the informed corrector also presents tradeoffs between quality of the denoiser and that of the corrector. While the HollowDiff scheme represents a compact algorithm for learning to generate and correct at the same time, practically, better performance is achieved by learning separate predictor and corrector models and combining them. Future work could investigate how this scheme can be made more efficient, potentially by architectural or algorithmic improvements on hollow transformers.

**Impact Statement.**  This paper aims to improve the efficiency and quality of discrete generative models, which have broad applications in areas such as image synthesis, text generation, and molecular design. Like many generative modeling techniques, the methods developed here could potentially be misused for generating misleading or harmful content (e.g., deepfakes or disinformation). We encourage responsible use and stress the importance of deploying generative models with appropriate safeguards and monitoring.

## Acknowledgements

This research was supported with Cloud TPUs from Google's TPU Research Cloud (TRC). Y.Z. was partially supported by the Stanford Interdisciplinary Graduate Fellowship (SIGF). S.D. and F.C. were partially supported by the McKnight Foundation, the Simons Foundation and an NIH CRCNS grant R01DC020874.

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

# A Diffusion Objectives

## A.1 Equivalence of objectives

For a complete picture, we include the derivation of the simplified objective in Shi et al. [12], Sahoo et al. [16], Ou et al. [17] and show how it connects to the Campbell et al. [8] objective expressed with CTMC transition rates. A similar treatment can be found in Shi et al. [12, App. H.1]. The derivation highlights that while the two expressions are mathematical equivalent, they nonetheless exhibit important differences in when it comes to estimation cost.

We start from Campbell et al. [8]'s continuous time discrete diffusion ELBO objective. For clarity, we use variable names $x$, $y$, and $z$ in a way that indicates temporal order:

$$\mathcal{L}(\theta) = \int \mathbb{E}_{q_t(y)r_t(z|y)} \left[ \left\{ \sum_{x' \neq y} \tilde{R}_t^\theta(y, x') \right\} - \left\{ \sum_{z' \neq y} R_t(y, z') \right\} \log(\tilde{R}_t^\theta(z, y)) \right] dt$$

$$= \int \mathbb{E}_{q_t(y)r_t(z|y)} \left[ \tilde{R}_t^\theta(y) - R_t(y) \log(\tilde{R}_t^\theta(z, y)) \right] dt, \tag{18}$$

where $R_t(y) \triangleq -R_t(y, y) = \sum_{z' \neq y} R_t(y, z')$. Here $r_t(z|y)$ represents the probability of transitioning from state $y$ to state $z$, given that we know a transition happens at time $t$:

$$r_t(z \mid y) = \begin{cases} \frac{R_t(y,z)}{R_t(y)} & y \neq z \\ 0 & y = z. \end{cases}$$

We can rewrite the objective with a label switching trick:

$$\mathcal{L}(\theta) = \int \mathbb{E}_{q_0(x_0)q_t(y|x_0)r_t(z|y)} \left[ \tilde{R}_t^\theta(y) - R_t(y) \log(\tilde{R}_t^\theta(z, y)) \right] dt$$

$$= \int \mathbb{E}_{q_0(x_0)q_t(y|x_0)} \left[ \tilde{R}_t^\theta(y) - \sum_{z \neq y} R_t(y, z) \log(\tilde{R}_t^\theta(z, y)) \right] dt$$

$$= \int \mathbb{E}_{q_0(x_0)q_t(y|x_0)} \left[ \sum_{z \neq y} \tilde{R}_t^\theta(y, z) - \sum_{z \neq y} R_t(y, z) \log(\tilde{R}_t^\theta(z, y)) \right] dt$$

$$= \int \mathbb{E}_{q_0(x_0)} \left[ \sum_y \sum_{z \neq y} \left\{ q_t(y \mid x_0) \tilde{R}_t^\theta(y, z) - q_t(y \mid x_0) R_t(y, z) \log(\tilde{R}_t^\theta(z, y)) \right\} \right] dt$$

$$= \int \mathbb{E}_{q_0(x_0)} \left[ \sum_y \sum_{z \neq y} \left\{ q_t(y \mid x_0) \tilde{R}_t^\theta(y, z) - q_t(z \mid x_0) R_t(z, y) \log(\tilde{R}_t^\theta(y, z)) \right\} \right] dt. \tag{19}$$

Note that we switched the labels $y$ and $z$ on the last line, which is the key trick that simplified our derivation. Proceeding from line (19) becomes straightforward:

$$\mathcal{L}(\theta) = \int \mathbb{E}_{q_0(x_0)q_t(y|x_0)} \left[ \sum_{z \neq y} \left\{ \tilde{R}_t^\theta(y, z) - R_t(z, y) \frac{q_t(z \mid x_0)}{q_t(y \mid x_0)} \log(\tilde{R}_t^\theta(y, z)) \right\} \right] dt$$

$$= \int \mathbb{E}_{q_0(x_0)q_t(y|x_0)} \left[ \sum_{x \neq y} \left\{ \tilde{R}_t^\theta(y, x) - \tilde{R}_{t|0}(y, x) \log(\tilde{R}_t^\theta(y, x)) \right\} \right] dt, \tag{20}$$

where $\tilde{R}_{t|0}(y, x) \equiv R_t(x, y) \frac{q_t(x|x_0)}{q_t(y|x_0)}$ is the true reverse transition rate conditioned on $x_0$, and we renamed $z$ to $x$ to preserve alphabetical order of the variables for readability. Now we have a simple Monte Carlo estimator for this objective: we just need to sample $y$ from the marginal distribution and then evaluate the parameterized transition model $\tilde{R}_t^\theta(y, x)$ for all neighbors $x$ of $y$, which can be done using only one model evaluation.

## A.2 Simplifying the Score Function for Masked Diffusion

The relationship between the (concrete) score function and denoising distribution can be simplified in the case of masked diffusion. We first start with the relation between the denoising distribution and score function in [8]:

$$s_t(y)_x = \frac{q_t(x)}{q_t(y)} = \sum_{x_0} \frac{q_{t|0}(x \mid x_0)}{q_{t|0}(y \mid x_0)} q_{0|t}(x_0 \mid y). \tag{21}$$

For a state $x$ where $x^d \neq \text{MASK}$, consider the score from $M^d(x)$ to $x$:

$$s_t^\theta(M^d(x))_x = \sum_{x_0} \frac{q_{t|0}(x \mid x_0)}{q_{t|0}(M^d(x) \mid x_0)} p_{0|t}^\theta(x_0 \mid M^d(x)) \tag{22}$$

$$= \sum_{x_0^d} \frac{q_{t|0}(x^d \mid x_0^d)}{q_{t|0}(\text{MASK} \mid x_0^d)} p_{0|t}^\theta(x_0^d \mid M^d(x)) \tag{23}$$

$$= \frac{q_{t|0}(x^d \mid x^d)}{q_{t|0}(\text{MASK} \mid x^d)} p_{0|t}^\theta(x^d \mid M^d(x)) \tag{24}$$

$$= \frac{\alpha_t}{1 - \alpha_t} p_{0|t}^\theta(x^d \mid M^d(x)) \tag{25}$$

where $\alpha_t$ represents the survival probability of any non-mask token at time $t$, which is assumed to be constant for all dimensions and token values. Going from (23) to (24) we used the fact that $x^d = x_0^d$ if $x^d \neq \text{MASK}$. This formula tells us that in the case of masking diffusion, the score function between mask and non-mask tokens is just a constant factor away from the denoising distribution.

In this work, we use a hollow transformer $f^\theta$ that outputs a $D \times S$ array, and we use $f^\theta(x, t)_{d,s}$ to represent $p_{0|t}^\theta(x^d = s \mid M^d(x))$. We can also omit the $t$ dependence of the network here because in absorbing state diffusion the true $q_{0|t}(x^d \mid M^d(x))$ does not depend on $t$ (proof omitted).

## A.3 Deriving the HollowDiff Objective

The hollow transformer provides us with a way to optimize the optimize the Campbell et al. [8] objective (18) directly:

$$-\log p(x_0) \leq \mathcal{L}(\theta) = \int_0^T \mathbb{E}_{q_{t|0}(y|x_0) r_t(z|y)} \Big[ \tilde{R}_t^\theta(y) - R_t(y) \log(\tilde{R}_t^\theta(z, y)) \} \Big] \, \mathrm{d}t + C, \tag{26}$$

$$= \int_0^T \mathbb{E}_{q_{t|0}(y|x_0)} \Big[ \sum_{x \neq y} R_t(x, y) s^\theta(y)_x - \sum_{z \neq y} R_t(y, z) \log(\tilde{R}_t^\theta(z, y)) \} \Big] \, \mathrm{d}t + C \tag{27}$$

$$= \int_0^T \mathbb{E}_{q_{t|0}(y|x_0)} \Big[ \sum_{x \neq y} R_t(x, y) s^\theta(y)_x - \sum_{z \neq y} R_t(y, z) \log(s_t^\theta(z)_y) \} \Big] \, \mathrm{d}t + C. \tag{28}$$

Notice that we can further simplify the second term by leveraging the sparsity structure of the forward process:

$$\sum_{z \neq y} R_t(y, z) \log(s_t^\theta(z)_y) = \sum_{d \in \overline{\mathcal{M}}(y)} R_t(y, M^d(y)) \log(s_t^\theta(M^d(y))_y), \tag{29}$$

$$= \sum_{d \in \overline{\mathcal{M}}(y)} R_t(y, M^d(y)) \log(\frac{\alpha_t}{1 - \alpha_t} f^\theta(y)_{d,y^d}), \tag{30}$$

$$= -\frac{\alpha_t'}{\alpha_t} \sum_{d \in \overline{\mathcal{M}}(y)} \log(f^\theta(y)_{d,y^d}) + C, \tag{31}$$

where $\alpha_t'$ represent the time derivative of $\alpha_t$. Note that here we used the hollow property $f^\theta(y)_d = f^\theta(M^d(y))_d$, which allows us to evaluate the sum over $d \in \overline{\mathcal{M}}(y)$ using only one function evaluation

of $f$. Similarly we can write out the first term, which conveniently reduces to a constant due to the structure of the forward process:

$$\sum_{x \neq y} R_t(x,y) s^\theta(y)_x = \frac{\alpha_t}{1 - \alpha_t} \sum_{d \in \mathcal{M}(y)} \sum_{\substack{x : x^d \neq \text{MASK}, \\ x^{\setminus d} = y^{\setminus d}}} R_t(x,y) f^\theta(y)_{d, x^d} \tag{32}$$

$$= -\frac{\alpha'_t}{1 - \alpha_t} \sum_{d \in \mathcal{M}(y)} \sum_{\substack{x : x^d \neq \text{MASK}, \\ x^{\setminus d} = y^{\setminus d}}} f^\theta(y)_{d, x^d} = -\frac{\alpha'_t}{1 - \alpha_t} |\mathcal{M}(y)|. \tag{33}$$

Here we are using the fact that $f^\theta(y)_d$ is a vector representing the distribution over the non-mask tokens and therefore sums to 1. Putting these back together, we have:

$$\mathcal{L}(\theta) = \int_0^T \frac{\alpha'_t}{\alpha_t} \mathbb{E}_{q_{t|0}(y|x_0)} \Big[ \sum_{d \in \mathcal{M}(y)} \log(f^\theta(y)_{d, y^d}) \} \Big] \, \mathrm{d}t + C, \tag{34}$$

which is very similar to the MD4 loss (5) of Shi et al. [12], except that the sum over all mask positions is replaced with the sum over all *non-mask* positions and the coefficients are different.

In practice, we want to maximize learning signals for the model. Since the objective (34) only uses the non-mask dimensions of $y$, we propose to combine it with the MD4 objective which uses the mask dimensions instead. These two simplified objectives are mathematically equivalent and have the same optima , but can suggest different stochastic gradient estimates. The combined objective is then

$$\mathcal{L}(\theta) = \int_0^T \frac{1}{2} \mathbb{E}_{q_{t|0}(y|x_0)} \Big[ \frac{\alpha'_t}{\alpha_t} \sum_{d \in \overline{\mathcal{M}}(y)} \log(f^\theta(y)_{d, x_0^d}) + \frac{\alpha'_t}{1 - \alpha_t} \sum_{d \in \mathcal{M}(y)} \log(f^\theta(y)_{d, x_0^d}) \} \Big] \, \mathrm{d}t + C,$$
$$\tag{35}$$

where we have changed the index in the first term from $y^d$ to $x_0^d$ for consistency, since $y^d = x_0^d$ when $d$ is an unmasked dimension. Note that the HollowDiff objective leads to a lower-variance stochastic gradient estimator, as it leverages more learning signal for a single sample of $y$. We confirm this empirically via loss curves in appendix D.2.

## B Informed Corrector

We explore informed corrector updates that leverages the hollow transformer architecture. Specifically, we consider the Gibbs operator:

$$q_t(x^d \mid x^{\setminus d}) = \sum_{x_0^d} q_{t|0}(x^d \mid x_0^d) q_{0|t}(x_0^d \mid x^{\setminus d}) = \alpha_t q_{0|t}(x^d \mid x^{\setminus d}) \mathbb{1}_{\{x^d \neq \text{MASK}\}} + (1 - \alpha_t) \mathbb{1}_{\{x^d = \text{MASK}\}}.$$
$$\tag{36}$$

To approximate this operator, we can use the learned network $f^\theta(x)_{d,i}$ in the place of $q_{0|t}(x^d = i \mid x^{\setminus d})$, since the hollow transformer architecture blocks out information of $x^d$ on the $d$th output dimension.

In order to reach the stationary distribution $q_t(x)$, we need to repeat the Gibbs update, iterating over all dimensions $d$. This is very inefficient in realistic settings. Instead, we choose to prioritize updates on the "most unconfident" dimensions, where $x^d$ is most likely to be affected by the compounding errors in the simulation of the backward process. To achieve this, we rank the dimensions according to the confidence score $c_d$ and perform $k$ updates at the same time.

One observation is that changing the mask configuration does not actually do anything useful for us, since the mask configuration does not contain any information. This suggests that we should instead consider the distribution conditioning on the mask configuration:

$$q_t(x^d \mid x^{\setminus d}, x^d \neq \text{MASK}) = \sum_{x_0^d} q_{t|0}(x^d \mid x_0^d, x^d \neq \text{MASK}) q_{0|t}(x_0^d \mid x^{\setminus d}, x^d \neq \text{MASK}) \tag{37}$$

$$= \sum_{x_0^d} \mathbb{1}_{\{x^d \neq \text{MASK}\}} \mathbb{1}_{\{x^d = x_0^d\}} q_{0|t}(x_0^d \mid x^{\setminus d}) \tag{38}$$

$$= \mathbb{1}_{\{x^d \neq \text{MASK}\}} q_{0|t}(x^d \mid x^{\setminus d}). \tag{39}$$

Repeatedly applying this operator on the non-masked dimensions creates a Markov chain with the stationary distribution equal to $q_t(x \mid \mathcal{M}_t) \triangleq q_t(x \mid x^{\mathcal{M}_t} = \mathrm{MASK})$ where $\mathcal{M}_t$ is the set of masked indices after the predictor update.

This translates to a straightforward procedure: find the non-mask dimensions with the lowest scores, and then sample from the model predictions $p_{0|t}^\theta(x^d \mid x^{\backslash d})$ of that dimension. For real-world datasets, we found that doing an argmax update in the end, where all dimensions are updated to the token that maximizes the conditional likelihood can be helpful for sample quality. We set the number of correctors $C = 1$ for the majority of our experiments. The algorithm is outlined in algorithm 1 below.

---

**Algorithm 1** Backward process with informed corrector steps

---

**Require:** $f, t_{min}, \theta$, temperature $\tau$;
**Require:** number of updates $k$ , number of predictor steps $P$, number of corrector steps $C$.
 1: Initialize time $t \leftarrow 1$
 2: Initialize sample $x \leftarrow \mathrm{MASK}^D$
 3: Set predictor step size $\Delta t \leftarrow (t - t_{min})/P$
 4: **for** step $= 1$ to $P$ **do**
 5:     Compute denoising probability $p[d, i] = f^\theta(x)[d, i]$
 6:     Apply predictor step $x \leftarrow \mathrm{PREDICTORUPDATE}(x, p, t)$
 7:     Update time $t \leftarrow t - \Delta t$
 8:     **for** $cstep = 1$ to $C$ **do**
 9:         Find most likely alternative tokens $x'^d = \arg\max_{x'^d \neq x^d} \log p_{0|t}^\theta(x'^d \mid x^{\backslash d})$
10:         Compute confidence score $c[d] = \log p_{0|t}^\theta(x^d \mid x^{\backslash d}) - \log p_{0|t}^\theta(x'^d \mid x^{\backslash d}), \forall d \notin \mathcal{M}(x)$
11:         Sample Gumbel noise $g[d] \sim \mathrm{Gumbel}(0, 1)$
12:         Compute ranking criterion $r[d] = -c[d] + \tau g[d]$
13:         // Update the top-k dimensions
14:         **for** $cnt = 1$ to $k$ **do**
15:             Get dimension of transition $d^* = \mathrm{SORTED}(r)[D - cnt]$
16:             Update state $x^{d^*} \leftarrow \mathrm{Categorical}(p_{0|t}^\theta(x^{d^*} \mid x^{\backslash d^*}))$
17:         **end for**
18:     **end for**
19: **end for**
20: // Deterministic update at the last step
21: Find most likely values for each mask dimension $x^d \leftarrow \mathrm{argmax}_{x_0^d} p_{0|t}^\theta(x_0^d \mid x)$ for all $d \in \mathcal{M}(x)$
22: **Return** $x$

---

## C   Hollow Transformer Architecture

### C.1   Hollow Transformer Architecture Details

We build upon the hollow transformer in [33] and design a class of hollow transformers that is more parameter efficient and expressive. First, we divide the computation into two streams, the content (causal) stream and the query (mixing) stream. The content stream consists of two transformers with causal self-attention in opposite directions, whereas the query stream attend to the content stream and outputs the final result. By design, non of the attention mechanisms allow a token at position $d$ to attend to a token that has access to information of $x^d$.

We differ from the original hollow transformer in that we introduce a third stream to combine information of the forward/backward causal streams. Specifically, we have the attention layers $F_l$, $B_l$, $M_k$ that implement multi-head attention with different causal structures. The updates of these attention streams are respectively (ignoring multilayer perceptron layers):

$$F_{l+1}^d \leftarrow \mathrm{Attention}(Q = F_l^d, KV = F_l^{\leq d}), \tag{40}$$

$$B_{l+1}^d \leftarrow \mathrm{Attention}(Q = B_l^d, KV = B_l^{\geq d}), \tag{41}$$

$$M_{k+1}^d \leftarrow \mathrm{Attention}(Q = h(M_k^d, F_{km}^d, B_{km}^d), KV = (F_{km}^{\leq d}, B_{km}^{\geq d})), \tag{42}$$

where we insert a mixing layer for every $m$ forward/backward layers. Notably, the new mixing layer $M_k$ acts like a "query stream" while the forward and backward layers $F_l$ and $B_l$ act like "content streams" [45]. To prevent information leak and preserve the "hollowness," we offset the inputs by one position when initializing the forward/backward layers:

$$F_1^d = \text{Embed}(\text{Pad}(x^{d-1})), \ B_1^d = \text{Embed}(\text{Pad}(x^{d+1})), \tag{43}$$

and we initialize the mixing layer to zero. Due to the attention structure, each position $M^d$ of the mixing layer can have access to all the other positions $x^{\backslash d}$ in the input $x$, and therefore it cannot attend to itself at other positions to preserve hollowness. We double the feature dimension of the mixing layer and set the update function $h$ to:

$$h(M_k^d, F_{km}^d, B_{km}^d) = M_k^d + \text{Concat}(F_{km}^d, B_{km}^d). \tag{44}$$

Finally, we use the output from the last mixing layer to compute the output logits.

**Weight tying.** Since the forward and backward content streams carry out the same type of computation, we use a weight-tied network where all weights are shared between $F_l$ and $B_l$, which significantly decreases the parameter count of our network and makes training and evaluation faster. In practice, we have not observed significant downgrade in performance of the weight-tied network on ImageNet.

## D  Experiments

The main experiments in this paper are performed on a v3-128 TPU pod and a v3-8 TPU machine with Google cloud.

### D.1  Hidden Markov Modeling

#### D.1.1  Hyperparameter Selection

We select the hyperparameters for the correctors in the Markov chain experiments with grid search. For the informed corrector, the number of parallel updates $k$ is swept over the values $\{1, 2, 4, 8, 16\}$, and the temperature hyperparameter is swept over $\{0.01, .1, .5, 1., 2., 4.\}$. For the uninformed corrector, the corrector step size is swept over $\{0.01, 0.1, 0.5, 1.0, 1.5, 2.0, 3.0, 4.0, 5.0\}$.

### D.2  Text8

The experiments on the Text8 dataset were performed on a machine with 8 NVIDIA H100 GPUs .

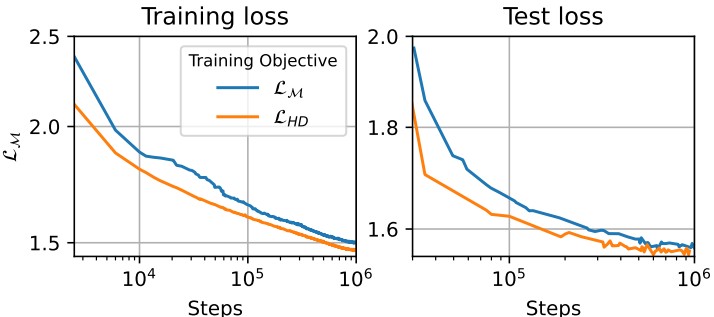

Figure 5: Loss curve on Text8 dataset [44]

We followed the standard dataset split and trained our models on text chunks of length 256 for 1 million steps with batch size 512. We train two models: one with the masked loss $\mathcal{L}_{\mathcal{M}}$ and the other with HollowDiff loss $\mathcal{L}_{\text{HD}}$ on the text8 dataset of Mahoney [44]. The model consists of 12 standard transformer layers, and we add one mixing layer at the end to combine the forward and backward streams. We empirically confirmed that the model trained with our proposed HollowDiff loss function $\mathcal{L}_{\text{HD}}$ is more efficient (see fig. 5).

### D.2.1 Example samples

We show some generated samples taken using informed correctors and MD4 in table 3.

Table 3: Comparison of unconditional samples from informed corrector and MD4 trained on Text8. Samples are generated with 256 NFEs. Words not contained in the training set vocabulary are marked in red.

| Informed corrector | MD4 |
|---|---|
| n of european population cities in northern indonesia the german integration of the societies in the united states the expansion of the province with the public education reform of one eight zero three the kingdom was held in one eight three five by the pe e two nine one zero one zero three zero two two four seven one zero zero three one three three five one one six four of the european union or trade union for it is the leniency of the netherlands five seven seven germany and three of the country s two five canadian publishing corp was created in one nine eight seven and helped to use it two days later it was designed or investigated according to the databases the date was also called out of the country in the one nine five zero s in the one nine six zero s i acturers of the developers rather than the starting elements of the day there are some useful models that restriction practice can be used as the standard command to give the same instruction this practice will make them a faithful with a complete list of | a deco dsssom d soylently taped for several meter blue for a time zone kutrata s mattom tte s space gas one nine five six the celliba or deitylic cylinder head designed by the longer gpu a thergameen ard glase one nine five four one nine four nine auto gav feature technology to harmonize anonymous aapproaches ragnar trifls livre by the word silva dam meaning small top or multi release good repodentiam defense genius multi pllo or hella spasca is derived from the verb meaning kill haltenere nickname d me mean ccording to naldi and closere clinton are a lucid living scientist of palger political views held into persist unlike high leg sacieties in sunghi s future which muscle meat low ly no chef is a public catalysis of the a frequent hebrid tlf vampires nympheo ic other area two km tho west of shire mael nontionment home of the united planets ampstone six zero one km long in five zero five ft two zero five kh total ground opening gin ashers air force navy the nine seven ford mk one five two mk three fa mz four fi |

## D.3 Tokenized ImageNet

### D.3.1 Training Details

Our model is adapted from that proposed by Sun et al. [33] and consists of 24 standard transformer layers. We use an embedding dimension of 784 and hidden dimension of 2048. We add 3 mixture layers (1 after every 8 transformer layers) to the model to facilitate information exchange between the forward and backward streams. We train the model using AdamW [46] with an initial learning rate of 1e-4 and a batch size of 512. We use a linear learning rate warmup in the first 100 steps. We adopt a stepwise learning rate schedule, dropping the learning rate to 3.3e-5 at $2.11 \times 10^6$ steps and to 1e-5 at $2.4 \times 10^6$ steps. We stop the training at $2.5 \times 10^6$ steps. The ImageNet experiments are done on v3-128 TPU pod machines on Google cloud and training takes 100 hours of wall clock time.

### D.3.2 Hyperparameter Selection

We select the hyperparameters for the samplers in the ImageNet experiments with grid search. For the informed corrector, the number of parallel updates $k$ is swept over the values $\{1, 2, 4, 8, 16\}$, and the temperature hyperparameter is swept over $\{.1, 1., 2., 10., 1000.\}$. For the uninformed corrector, the corrector step size is swept over $\{.5, 1., 2., 4.\}$. For maskgit, the temperature hyperparameter is swept over $\{0.5, 1., 2., 4., 8., 10., 12., 16., 20., 40.\}$.

### D.3.3 Other Sampling Details

**Final argmax update.** We found empirically that applying a final update of the form

$$\hat{x}_0^d = \operatorname*{argmax}_{x_0^d} \ p_{0|t}^{\theta}(x_0^d \mid \hat{x}_t^{\setminus d}) \tag{45}$$

for all $d \in \{1, \dots, D\}$ improves the results for the single network informed corrector experiments. Our intuition is that this final update helps eliminate remaining noise and local errors from all positions. When the base denoising model is stronger than the corrector model, as is the case when the informed corrector is applied on top of MD4 predictors, this special update becomes unnecessary.

### D.3.4 Ablation on confidence metrics.

We compared two confidence scoring schemes used in the informed corrector: the log-likelihood of the predicted token and the margin between the top-1 and top-2 logits. As shown below (appendix D.3.4), the margin-based confidence yields slightly better FID scores in both settings.

| Model | Confidence Type | FID $\downarrow$ |
|---|:---:|:---:|
| HollowDiff + informed corrector | log-likelihood | 6.45 |
| HollowDiff + informed corrector | margin | **6.26** |
| MD4 + informed corrector | log-likelihood | 5.85 |
| MD4 + informed corrector | margin | **5.78** |

Table 4: Comparison of log-likelihood vs margin-based confidence scores.

### D.3.5 Generated Image Examples

We show some ImageNet samples taken using HollowDiff with informed correctors in figs. 6 to 8.

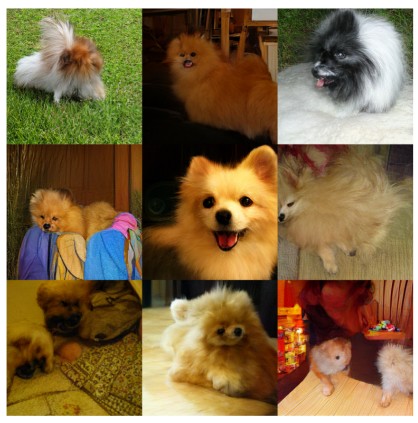

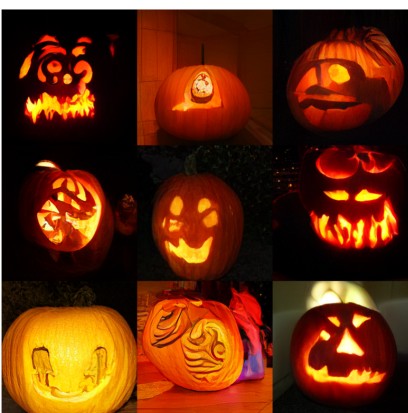

Figure 6: Example informed corrector generations for the classes Pomeranian (259) and jack-o'-lantern (607).

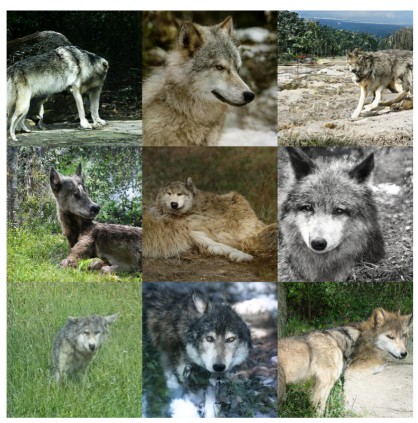

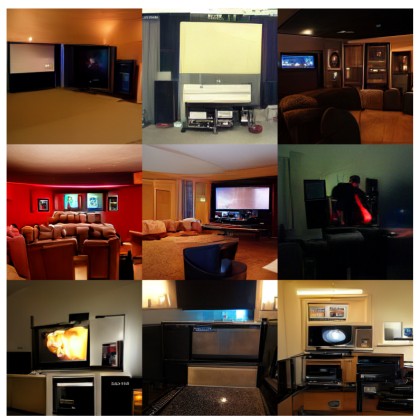

Figure 7: Example informed corrector generations for the classes timber wolf (269) and home theater (598).

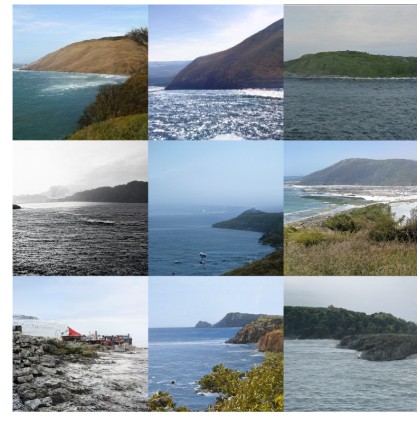

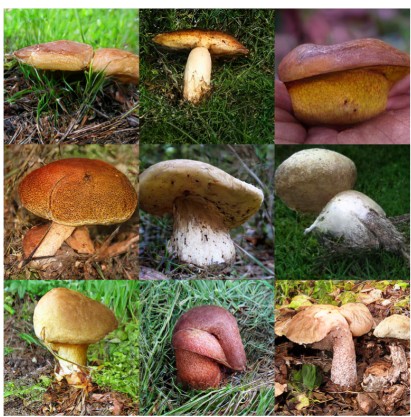

Figure 8: Example informed corrector generations for the classes seashore (978) and bolete (997).

