# OpenReview forum: "Informed Correctors for Discrete Diffusion Models"
_NeurIPS.cc/2025/Conference — NeurIPS 2025 poster_

### Official Review · Reviewer_88LA · 2025-06-30

**Clarity:** 2
**Significance:** 2
**Originality:** 3
**Rating:** 4
**Confidence:** 4

**Summary:**

The paper identifies a key inefficiency in traditional predictor-corrector sampling within mask absorbing diffusion models, specifically that it fails to effectively target problematic tokens. To address this, the authors propose a Gibbs sampling-based mask absorbing diffusion framework. The forward process is defined conditionally for each dimension based on the values of the other dimensions, enabling training through a redefined ELBO term. They introduce the "hollow transformer" architecture, specifically designed to handle this training regime. During sampling, the authors employ a Hogwild Gibbs sampling approach, selecting k dimensions based on their proposed confidence heuristic. The authors report improvements in sample quality as a result.

**Questions:**

Q1: What specific theoretical advantages does the proposed confidence heuristic offer compared to heuristics in existing literature? Although the Gibbs sampling-based mask absorbing diffusion framework is theoretically sound, its comparative advantage over existing frameworks should be also theoretically justified.

Q2: How does the proposed Gibbs sampling approach scale with respect to dimensionality and complexity in realistic datasets?

**Ethical Concerns:**

["NO or VERY MINOR ethics concerns only"]

**Final Justification:**

The authors' explanation addresses my questions, so the reviewer will maintain the original score, which is already positive.

**Limitations:**

yes

**Quality:**

2

**Strengths And Weaknesses:**

**Strengths**
The theoretical justification for the proposed Gibbs sampling-based diffusion framework, which builds upon existing CTMC frameworks, is robust. The idea of selectively correcting problematic dimensions to enhance sampling efficiency is intuitive and appears promising.

**Weaknesses**

1) Theoretical Perspective:  Although the Gibbs sampling-based mask absorbing diffusion framework is theoretically sound, its comparative advantage over existing frameworks remains unclear. The proposed confidence heuristic for informed correction seems lacking theoretical justification, raising doubts about its superiority over heuristics presented in previous methods such as LLADA.
2) Experimental Perspective: The experiments provided are limited in scope and scale, focusing only on an HMM-based toy experiment, character-level text modeling, and ImageNet (256x256). Important domains mentioned in the introduction, like language modeling and protein generation, lack experimental validation. Even within the character-level text modeling experiments, key metrics such as perplexity, standard in the NLP community, are not reported.

---

> ### Author Rebuttal · Authors · 2025-07-31
>
> We thank the reviewer for carefully considering our work and recognizing the theoretical soundness of our Gibbs sampling-based framework, as well as the promise of selectively correcting problematic dimensions. Below, we address the main concerns.
>
> ### **Q1: Theoretical basis for the confidence heuristic**
>
> We thank the reviewer for raising this important question. Although our adaptive Gibbs sampler depends on a confidence metric conceptually similar to those in literature [Nie et al. 2025, Kim et al. 2025], our method is unique and more principled. Other methods, like the one used in LLaDA, alter the predictor's unmasking order based on confidence. This can introduce bias into the sampling process, causing it to deviate from the diffusion model's theoretical reverse process. In contrast, our confidence metric is only used in corrector to define the selection distribution of a random-scan Gibbs sampler, which approximately preserves the marginal distributions of reverse process. Therefore, our sampling process continues to closely approximate the true reverse process of the diffusion model.
>
> We will clarify these motivations and connections and add the references in the final version.
>
> Nie, Shen, et al. "Large language diffusion models." *arXiv preprint arXiv:2502.09992* (2025).
>
> Kim, Jaeyeon, et al. "Train for the Worst, Plan for the Best: Understanding Token Ordering in Masked Diffusions." *Forty-second International Conference on Machine Learning*.
>
> ### **Q2: Scaling to high-dimensional data**
>
> Our informed corrector is specifically designed to scale well in high-dimensional settings. The hollow transformer architecture enables evaluating confidence scores for all dimensions in a single forward pass, and the Gumbel-top-k trick allows us to select and update multiple uncertain positions efficiently in parallel.
>
> For more discussion on this, refer to our response to Reviewer KG2Y on scaling to masked diffusion based large language models.
>
> ### **Experimental scope and evaluation metrics**
>
> We clarify that our informed corrector method focuses on improving discrete diffusion sampling, an aspect not captured by likelihood-based metrics such as perplexity. For this reason, our evaluation is based on sample quality metrics, namely word error rate (on text8) and FID (on ImageNet).
>
> Given our computational constraints, we prioritized demonstrating the generality of our method across diverse evaluation settings, including HMMs, character-level text, and images, rather than pushing for scale within each domain. We believe our our method is readily applicable to other discrete domains including protein and large-scale language modeling and we're excited to explore these applications in future work.

---

> ### Comment · Reviewer_88LA · 2025-08-04
>
> Thanks for the explanation. Your explanation addresses my questions, so the reviewer will maintain the original score, which is already positive.

---

> ### Comment · Area_Chair_S7yZ · 2025-08-04
>
> In light of the rebuttal and the other reviews/rebuttals, please edit this comment and update with reasoning for /why/ you keep the rating unchanged.

---

### Official Review · Reviewer_KG2Y · 2025-07-02

**Clarity:** 4
**Significance:** 4
**Originality:** 4
**Rating:** 5
**Confidence:** 5

**Summary:**

- Discrete diffusion models are an increasingly popular class of generative models for text, proteins, and images. Since the computational cost of each forward pass is high, most methods sample tokens in parallel, introducing errors in generation and reducing sample quality. The authors of this paper propose a predictor-corrector sampling scheme by learning a corrector model.
- The predictor-corrector scheme is based on adaptive Gibbs sampling which makes use of the corrector model proposed in the paper. Unlike typical Gibbs samplers, which sample one coordinate at a time, the authors make use of parallel updates which are then corrected using their corrector model.
- The corrector model makes use of a hollow transformer such that the corrector model does not take into account the position whose correctness is being judged, reducing the number of forward calls required to 1.
- The authors derive an ELBO for training the correct model which can be combined with the denoiser.
- As baselines, the authors compare against uninformed correctors such ReMDM which mask out text randomly or based on weights computed from previous denoising time-steps.

**Questions:**

See weaknesses

**Ethical Concerns:**

["NO or VERY MINOR ethics concerns only"]

**Final Justification:**

The authors have addressed my concerns and answered my questions. The work done here is original, interesting and timely.

**Limitations:**

yes

**Quality:**

4

**Strengths And Weaknesses:**

Strengths

- The proposed architecture for the corrector model allows efficient inference.
- The proposed objective leads to faster training as shown in figure 5 in appendix D.2
- The adaptive Gibbs sampler with parallel updates and then correction leads to faster convergence across various NFEs.
- For MD4 table 1 does show improved performance with the informed corrector.

Weaknesses

- The experimental evaluations can be improved, for instance, the table 1 compares performance across different parameter counts.
- Several large-scale masked diffusion language models have been released earlier this year, it would be great to see if a separate corrector model can be scaled to larger parameter counts.

---

> ### Author Rebuttal · Authors · 2025-07-31
>
> We thank the reviewer for the positive and thoughtful assessment. We address the weaknesses identified by the reviewer below.
>
> ### **Parameter counts in Table 1**
>
> We appreciate the reviewer’s observation regarding parameter mismatches in Table 1. Given the breadth of existing work, it is computationally infeasible to reproduce and fairly re-train every baseline under a fixed parameter budget. As such, we chose to focus our evaluation on comparisons with MD4 to better isolate and demonstrate the benefits of the informed corrector. Other entries such as DPC and VQ-Diffusion are included to provide broader context on the landscape of discrete diffusion models.
>
> We also note that the hybrid MD4 + informed corrector setup uses more total parameters (400M) than the unified HollowDiff model (230M). This comparison is not intended to be strictly fair, but rather to highlight the modularity and compatibility of our corrector: it can be applied to stronger pretrained predictors without need for joint training. We will clarify these distinctions in the final version of the paper.
>
> ### **Scaling to larger models**
>
> We agree that scaling to larger masked diffusion models is a promising and important direction. Our current experiments use relatively lightweight models in order to isolate and rigorously evaluate the contribution of the corrector scheme.
>
> That said, we are optimistic about the scalability of informed correctors, particularly in the context of modern language models, where output sequences are much longer and generation speed is a key concern. As longer outputs are generated in fewer steps, more tokens are sampled in parallel, increasing the risk of errors. This makes the ability to perform efficient, parallel, and targeted corrections even more valuable. We believe informed correctors are well-positioned to address this need.
>
> We leave this direction to future work.

---

> ### Comment · Area_Chair_S7yZ · 2025-08-04
>
> Reviewer KG2Y, please also engage in discussion with an official comment and not just clicking the mandatory acknowledgement button. Any follow-up questions? Reflections on why the rebuttal changed your opinion on the submission? Or why it did not?

---

### Official Review · Reviewer_PFik · 2025-07-03

**Clarity:** 3
**Significance:** 2
**Originality:** 2
**Rating:** 4
**Confidence:** 3

**Summary:**

This paper proposes a novel predictor-corrector sampling scheme for discrete diffusion models to improve the efficiency of inference process.
The technique proposed is an informed corrector that leverages the diffusion model itself to identify and resample tokens that are most likely to be erroneous, drawing inspiration from adaptive Gibbs sampling.
Specifically, to quantify the model's certainty, it constructs a
confidence score for each generated token.
This score is defined as the margin between the logit of the current token and that of the next most likely alternative.
To efficiently enable the informed corrector, the authors employ a hollow transformer architecture for parallel processing with a single forward pass and modify the training objective that combines complementary loss terms to improve training efficiency.
In experiments, the authors conduct synthetic text and image generation tasks, the proposed informed corrector consistently outperforms baselines including predictor-only samplers, the uninformed corrector, and the ReMDM sampler.

**Questions:**

As shown in Figure 4, the FID score rises sharply when the number of parallel corrector updates $k$, becomes too large (e.g., k ≥ 32). Could the authors provide a more intuitive explanation for why excessive parallel updates harm sample quality.

**Ethical Concerns:**

["NO or VERY MINOR ethics concerns only"]

**Final Justification:**

The authors have addressed my previous concerns. I am now prone to accept the manuscript.

**Limitations:**

yes

**Quality:**

3

**Strengths And Weaknesses:**

### Strength
- The proposed method is natural and intuitive, introducing a well-motivated informed corrector that directly targets likely errors.
This marks a significant improvement over prior uninformed methods, which rely on inefficient random re-masking.
- The use of the hollow transformer architecture is a natural way to efficiently implement the parallel updates required by the informed corrector.

### Weaknesses
The core idea is easy to follow and insightful, several aspects could benefit from further clarification and investigation:
- The paper compares performance based on the NFEs but does not report wall-clock sampling times and the memory usages. Since a corrector step includes confidence score calculation and resampling, a time-based comparison would offer a more complete picture of the method's practical efficiency.
- In the Discussion, the authors state that "better performance is achieved by learning separate predictor and corrector models". While Table 2 shows the superior result of this informed corrector, could the authors provide a more direct comparison to quantify the performance difference between the jointly trained models and the separately trained models?

---

> ### Author Rebuttal · Authors · 2025-07-31
>
> We thank the reviewer for the thoughtful summary and for recognizing the clarity, motivation, and practical significance of our proposed informed corrector and hollow transformer architecture. We respond below to the raised concerns and questions.
>
> ### **Wall-clock time comparison**
>
> To show that a corrector step does not significantly increase the wall-clock time of a sampling step, we compare the sampling time of informed corrector (HollowDiff) with a HollowDiff without corrector, we can see that additional time taken by the corrector step is negligible. However, we did notice that our Hollow Transformer implementation currently has a slightly slower forward pass than a regular Transformer, as shown in the table.
>
> In the table, wall clock time in seconds is measured for generating sequences of  256 tokens with a batch size 64 on a machine with 8 NVIDIA H100 SXM GPUs. The first row indicates the number of NFEs.
>
> |  | 16 | 32 | 64 | 128 | 256 |
> | --- | --- | --- | --- | --- | --- |
> | Informed Corrector | 0.1120 ± 0.0003 (k=16) | 0.2098 ± 0.0005 (k=8) | 0.4052 ± 0.0003 (k=4) | 0.7965 ± 0.0004 (k=1) | 1.5775 ± 0.0001 (k=1) |
> | Hollow Transformer (no corrector) | 0.1100 ± 0.0001 | 0.2092 ± 0.0001 | 0.4061 ± 0.0003 | 0.8003 ± 0.0002 | 1.5885 ± 0.0001 |
> | md4 (uninformed corrector) | 0.0831 ± 0.0004 | 0.1343 ± 0.0006 | 0.2349 ± 0.0008 | 0.4260 ± 0.0005 | 0.8242 ± 0.0007 |
> | md4 (ReMDM) | 0.0684 ± 0.0004 | 0.1214 ± 0.0005 | 0.2319 ± 0.0003 | 0.4489 ± 0.0003 | 0.8928 ± 0.0003 |
> | md4 (no corrector) | 0.0618 ± 0.0008 | 0.11160 ± 0.00008 | 0.2127 ± 0.0004 | 0.4154 ± 0.0002 | 0.8198 ± 0.0003 |
>
> While the hollow transformer does incur a slowdown, we note that we did not explicitly optimize the computation graph and the architecture for wall-clock time. We are confident that further optimizations will significantly decrease this gap in performance.
>
> ### **Joint vs. separate predictor/corrector models**
>
> You are correct that the best performance in Table 2 comes from the decoupled setup (MD4 predictor + hollow corrector, FID 5.78). However, this configuration uses more total parameters (400M vs. 230M for HollowDiff) and thus is not intended as a strictly fair comparison. Rather, we include it to demonstrate that the informed corrector generalizes well to stronger predictors and is not tied to a particular architecture.This is intended as a proof of concept, and we are excited to explore such hybrid modeling opportunities in future work.
>
> ### **Why excessive parallel updates hurt performance (Figure 4)**
>
> We appreciate the reviewer’s insightful question. This is indeed an important point to clarify.
>
> The informed corrector is theoretically grounded in Gibbs sampling, and the correspondence is exact in the limit of single-coordinate updates (k = 1) and many corrector steps. As k increases, we improve the chance of hitting generation errors in fewer steps, but we also move away from the Gibbs framework.
>
> Intuitively, each update assumes the rest of the sequence remains fixed during correction. When too many dimensions are updated in parallel, this assumption no longer holds. The result is a kind of “update collision” — simultaneous updates may become mutually inconsistent, or even undo each other’s corrections. This degrades the effectiveness of the corrector and ultimately harms sample quality. The ablation results on k tells us that there is a good middle ground where some number of parallel updates allow faster generation without hurting performance. We will clarify this point in the revised caption for Figure 4.

---

> > ### Comment · Area_Chair_S7yZ · 2025-08-05
> >
> > Dear reviewer PFik,
> >
> > After considering the rebuttal to your review and the other reviews/rebuttals, how and why has this affected your position on this submission? Please reply with an official comment (not just the mandatory acknowledgement) reflecting your current view, any follow-up questions/comments etc.
> >
> > Note the Aug 6 AoE deadline, make sure to respond in time for the authors to be able to submit a response if necessary.

---

### Official Review · Reviewer_E3jH · 2025-07-04

**Clarity:** 3
**Significance:** 3
**Originality:** 2
**Rating:** 4
**Confidence:** 3

**Summary:**

This paper proposes a novel informed corrector for discrete diffusion models, aimed at improving sample quality and efficiency during inference. While discrete diffusion has shown promise across a range of tasks—such as language and image generation—the sampling process remains computationally expensive and prone to errors when using fewer steps. Existing uninformed correctors make naive, often random, corrections without considering model uncertainty, leading to suboptimal performance. To address these issues, the authors introduce: 1. An informed corrector based on adaptive Gibbs sampling, which targets uncertain or likely incorrect dimensions using a confidence-based criterion. 2. A hollow transformer architecture that allows efficient multi-token updates without information leakage. 3. A complementary ELBO-based training objective, named HollowDiff, which leverages both masked and unmasked positions for more efficient learning. Experimental results validate the effectiveness of the proposed method.

**Questions:**

1. Could the authors provide a direct ablation comparing the two proposed confidence scores (log-likelihood vs. margin-based)? This would clarify their individual contributions.
2. The corrector is only activated when t < 0.9. How sensitive is this threshold, and would more frequent corrector steps improve performance or introduce instability?
3. While the hollow transformer enables efficient parallel updates, it underperforms compared to a standard transformer in denoising tasks (Table 2). Additionally, the proposed informed corrector demonstrates greater effectiveness when used with the hollow transformer. Could the authors elaborate on potential architectural improvements to reduce this performance gap while preserving the corrector’s efficiency benefits?

**Ethical Concerns:**

["NO or VERY MINOR ethics concerns only"]

**Final Justification:**

The reviewer addressed my concerns. Considering the contribution and novelty of the paper, I will maintain my original score of slightly positive.

**Limitations:**

see above

**Quality:**

3

**Strengths And Weaknesses:**

Strengths:
1.The adaptive corrector design is a clear advancement over uninformed methods and incorporates techniques (e.g., Gumbel-top-k) not previously applied in this context.
2. The design of the corrector is both novel and practically effective, with consistent empirical improvements.
3,The writing is clear and logically structured.

Weakness:
1. The requirements on separately trained predictor and corrector models for best performance adds complexity to the training pipeline, somewhat diminishing the elegance of the proposed unified approach.
2. The hollow transformer, while clever, imposes a constraint that makes it incompatible with arbitrary pretrained denoisers. This limits plug-and-play applicability.
3. Some architectural choices  could benefit from more ablation or sensitivity analysis.

---

> ### Author Rebuttal · Authors · 2025-07-31
>
> We thank the reviewer for the constructive and thoughtful feedback. We are pleased that you found our method novel, effective, and clearly written. Below we respond to your questions and concerns.
>
> ### **Ablation: log-likelihood vs. margin-based confidence scores**
>
> We agree with the reviewer that the ablation on the confidence score is an important result to include. We give the results in the table below. All numbers are reported with 17 NFEs with 8 corrector steps.
>
> | Method | Confidence | FID |
> | --- | --- | --- |
> | HollowDiff + informed corrector | log-likelihood | 6.45 |
> | HollowDiff + informed corrector | margin | **6.26** |
> | MD4 + informed corrector | log-likelihood | 5.85 |
> | MD4 + informed corrector | margin | **5.78** |
>
> It can be seen that margin-based confidence yields better performance. We will include this table in the appendix for the final version.
>
> ### **Sensitivity of the corrector activation time**
>
> Thank you for pointing this out. In earlier versions of our implementation, we applied a threshold to delay the start of corrector updates (e.g., t < 0.9) following [8]. However, in our final experiments, we found this unnecessary and removed the threshold entirely. We regret the oversight in leaving this outdated detail in the appendix and will correct it in the final version.
>
> ### **Hollow Transformer underperformance and potential improvements**
>
> We agree with the reviewer’s observation that the hollow transformer underperforms compared to a standard transformer in denoising tasks. We believe that hollow transformers are difficult to train because it is designed to simultaneously learn both the predictor and the corrector via a structural constraint. While there should be a fair amount of weight sharing between the two learning tasks, it is fundamentally a harder learning problem than learning the predictor (denoising) on its own.
>
> One potential way to speed up training and also address the incompatibility of hollow transformers is by fine-tuning a “hollowed out” pretrained denoising transformer, using its learned weights to initialize the layers of a hollow transformer. We will leave this line of inquiry for future work.
>
> ### **On unified vs. separate training**
>
> We agree that requiring separate predictor and corrector models adds complexity. However, we view this not as a limitation but as flexibility. Our unified HollowDiff model (230M params) already performs competitively (FID 6.26). The hybrid setup (MD4 + hollow corrector) achieves even better performance (FID 5.78), but it uses more parameters (400M total). We do not claim this as a strictly fair comparison to unified approaches — rather, we include it to demonstrate that our informed corrector is broadly compatible and effective even when paired with stronger pretrained predictors. We will clarify this point in the final version.

---

> > ### Comment · Reviewer_E3jH · 2025-08-04
> > **response**
> >
> > Thank you for your response. The authors have addressed my concerns well, and I will keep my original score.

---

> > > ### Comment · Area_Chair_S7yZ · 2025-08-05
> > >
> > > Dear reviewer E3jH,
> > >
> > > Please update the response with a motivation on why the original score reflects your current updated position.

---

### Note · Authors · 2025-08-15

We sincerely thank the AC for handling the submission and the reviewers for their thoughtful and constructive feedback. We are pleased that the reviewers found our core contribution — the informed corrector — along with the use of hollow transformer and redefined ELBO objective to be novel (E3jH, PFik), intuitive (88LA) and practically effective (E3jH). We also appreciate the recognition of our method’s sampling improvements and its potential to generalize across modalities.

**Addressing the reviewers comments:** we clarified the motivation and tradeoffs of using parallel updates in the corrector, and explained the degradation in sample quality at high values of *k* through the lens of update collisions. Regarding model capacity, we clarified that the comparison between MD4 (230M) and MD4 + informed corrector (400M) was intended to demonstrate the modularity of our method and its compatibility with pretrained denoisers. We also elaborated on experimental details, such as wall-clock performance and ablations across different confidence score schemes.

On the theoretical design of our method, we also provided our perspective on how our informed corrector (based on adaptive Gibbs sampling) compares against other recent sampling heuristics (e.g. LLaDA). We highlighted our approach’s soundness in preserving the marginal distribution under the diffusion framework. We also discussed the scaling potential of informed correctors, particularly for long-sequence generation tasks, where parallelism is of great practical importance.

We are grateful for the reviewers’ insights and hope our clarifications strengthen the case for this work.

---

### Decision · Program_Chairs · 2025-09-17

**Decision:**

Accept (poster)

**Comment:**

The submission introduces informed correctors for sampling discrete diffusion models. The approach leverages adaptive Markov chains to enable more efficient and accurate sampling. While it's unclear whether the stationary distribution remains the same for the given adaptive selection and hogwild update of indices, the method seems to perform well in practice.

The reviewers reached a consensus recommending acceptance based on a novel application of Gibbs sampling to diffusion models, improved efficiency, intuitive/timely approach. After considerations of the points I see no reason to recommend differently. Please make sure to include the additional experiments, clarifications and additional discussions/points raised by the reviewers in the revision.